# Genetic, Phenotypic, and Clinical Heterogeneity of NPM1-Mutant Acute Myeloid Leukemias

**DOI:** 10.3390/biomedicines11071805

**Published:** 2023-06-24

**Authors:** Ugo Testa, Elvira Pelosi, Germana Castelli

**Affiliations:** Department of Oncology, Istituto Superiore di Sanità, Viale Regina Elena 299, 00161 Rome, Italy; elvira.pelosi@iss.it (E.P.); germana.castelli@iss.it (G.C.)

**Keywords:** acute myeloid leukemia, genetic classification, mutational profiling, transcriptome analysis, prognostic stratification, clonal evolution

## Abstract

The current classification of acute myeloid leukemia (AML) relies largely on genomic alterations. AML with mutated nucleophosmin 1 (*NPM1-mut*) is the largest of the genetically defined groups, involving about 30% of adult AMLs and is currently recognized as a distinct entity in the actual AML classifications. *NPM1-mut* AML usually occurs in de novo AML and is associated predominantly with a normal karyotype and relatively favorable prognosis. However, *NPM1-mut* AMLs are genetically, transcriptionally, and phenotypically heterogeneous. Furthermore, *NPM1-mut* is a clinically heterogenous group. Recent studies have in part clarified the consistent heterogeneities of these AMLs and have strongly supported the need for an additional stratification aiming to improve the therapeutic response of the different subgroups of *NPM1-mut* AML patients.

## 1. Introduction

The development of the techniques for the analysis of genomes has greatly contributed to a detailed molecular classification of acute myeloid leukemia. In 2016, a first genomic classification of Acute Myeloid Leukemias (AMLs) was proposed that distinguishes 11 molecular subtypes, each with peculiar diagnostic molecular features [1]. More recently, according to the mutational profile and cytogenetic analysis, this classification was updated and revised, supporting the existence of 16 molecular classes [2]. These molecular classifications have a relevant role in the diagnosis and treatment of AML patients and have been included in recent internationally accepted systems of leukemic classification or risk stratification, such as the 2022 World Health Organization (WHO) classification [3], the new International Consensus Classification (ICC) [4], and the European Leukemia Net (ELN) risk stratification [5]. All of these classifications prioritize the role of genetic alterations to establish diagnosis and prognosis and to have criteria for the definition and evaluation of minimal residual disease, and, in some instances, the indication of the optimal treatment.

Mutations at the nucleophosmin 1 (*NPM1*) level represent one of the most common gene mutations (25–30% of cases) observed in adult AML patients [6]. *NPM1* encodes a multifunctional protein, prominently localized at the level of the nucleolus, that shuttles between the nucleus and cytoplasm; the mutant NPM1 protein is delocalized at the level of the cytoplasm [7]. The nuclear export is mediated by the interaction of the mutant NPM1 protein with exportin 1 (XPO1), a nuclear transporter acting as a direct carrier mediating the export of proteins containing a nuclear export signal into the cytoplasm. The biochemical and functional properties of normal and mutant NPM1 protein have been recently reviewed by Falini et al. [7]. Usually, *NPM1*-mutant AMLs at diagnosis display high percentages of blasts, high white cell counts, and increased extramedullary involvement and are commonly associated with a normal karyotype [7]. Only about 15% of these patients display an abnormal karyotype, with the most frequent chromosomal abnormalities being represented by +8, +4, del 9q, and +21 [7]. The screening of a large cohort of 2426 *NPM1*-mut AML patients negative for *FLT3-ITD* or with a low *FLT3-ITD* allelic ratio (*NPM1*-mut/*FLT3-ITD*^neg/low^) showed that 17.6% of these patients displayed an abnormal karyotype: 13.6% of patients had intermediate-risk and 3.4% had adverse-risk chromosomal abnormalities [8]. Overall survival and event-free survival were significantly reduced in patients with adverse-risk chromosomal abnormalities [8].

*NPM1* mutations are heterogeneous and mostly localized at the level of exon 12 of the *NPM1* gene [2]. *NPM1* mutations are always heterozygous and are caused by 4 bp insertions inducing a frameshift mutation at the C-terminus of the NPM1 protein resulting in a loss of tryptophan residues (w288 and W290 or W290 alone) and the gain of a new nuclear export signal (NES) determining a disruption of the folded helix structure with the loss of the nucleolar localization signal (NoLS): all of these changes determine a shift towards the nuclear export and cytoplasmic localization of the NPM1 protein [2]. These two tryptophan residues are responsible for the nucleolar localization signal and interaction with ribosomal DNA. According to the different types of *NPM1* mutations, *NPM1-mut* AMLs are subdivided into three main subgroups: type A, characterized as an insertion of TCTG between nucleotides 860 and 863 (69% of cases); type B, characterized by the insertion of CATG between nucleotides 863 and 864 (11% of cases); and type D, characterized by the insertion of CCTG between nucleotides 863 and 864 [9]. Type A *NPM1-mut* AMLs are characterized by a high frequency of *DNMT3A* mutations [9].

The effect of mutant *NPM1* is dominant over the normal *NPM1* allele, a phenomenon caused by the formation of heterodimers between mutant and normal *NPM1*, delocalized at the level of the cytoplasm [2]. The dominance of the mutant allele over the normal *NPM1* allele is also reinforced through the preferential transcription of the mutant allele [10].

The mechanisms through which *NPM1-mut* causes leukemic transformation remain undetermined. A key role seems to be played by the dysregulation of developmental and stem cell-associated genes such as *HOXA* cluster genes and *MEIS1*, highly expressed in *NPM1-mut* AMLs. The high expression of *HOX* genes in *NPM1-mut* AML cells requires the presence of the delocalized NPM1 mutant protein; in fact, pharmacological inhibition of the XPO1-relocalized NPM1 mutant protein in the nucleus results in the immediate downregulation of *HOX* gene expression, the differentiation of AML cells, and the prolongation of the survival of *NPM1-mut* leukemic mice [11]. Two recent studies have clarified the mechanism through which *NPM1-mut* directly upregulates the expression of target genes. Uckelmann et al. showed that *NPM1-mut* directly binds at the level of specific chromatin gene targets, co-occupied by the histone methyltransferase KMT2A (MLL1); the targeted degradation of NPM1 determines a rapid decrease in gene expression and in activating histone modifications at the level of target genes [12]. Wang et al. showed that NPM1-mut binds at the level of active gene promoters in *NPM1-mut* AML cells, including *HOXA/B* gene clusters and *MEIS1*; NPM1-mut sustains the transcriptional activation of these genes by inhibiting the activity of histone deacetylases [13]. Studies based on mouse models have shown that *NPM1* mutations favor leukemic transformation through a double mechanism: the hyperactivation of NPM1 target genes *MEIS1* and *HOXA* and the induction of a condition of haploinsufficiency of *NPM1*-WT determining an insufficient level of normal NPM1 protein at the level of the nucleus and nucleolus [14].

## 2. The Mutational Landscape of *NPM1*-Mutant AMLs

*NPM1*-mutant AMLs have been characterized in detail for their mutational profile, showing their frequent association with co-mutations. Some of these co-mutations play a key role in *NPM1*-mut AML development and are prognostically relevant.

Studies on large cohorts of *NPM1*-mut AMLs showed recurrent mutations of genes involved in DNA methylation (*DNMT3A* (51%), *TET2* (15.5%), *IDH1* (12%), *IDH2* (14%), and *WT1* (8%)) and activated signaling (*FLT3-ITD* (39.5), *NRAS* (19%), *FLT3-TKD* (17.5%), *PTPN11* (16%), *KRAS* (4%)) [15,16] (Figure 1). More than 95% of *NPM1*-mut AMLs display co-mutations of at least one of these genes [15,16]. The analysis of individual *NPM1*-mut AMLs showed that the large majority of cases with *DNMT3A* mutations display concomitant mutations of one or more than one gene of DNA methylation or activated signaling pathways: particularly frequent is the co-association with *FLT3-ITD*, *FLT3-TKD*, *TET2*, *IDH1*, *IDH2*, *WT1,* and *NRAS* mutations [15]. In other cases, *TET2*, *IDH1*, *IDH2*, *WT1*, *FLT3-ITD*, *FLT3-TKD,* and *NRAS* mutations are not associated with *DNMT3A* mutations; in these cases, *FLT3-ITD* mutations are frequently associated with *IDH2*, *WT1,* and *TET2* mutations, while *FLT3-TKD* mutations are frequently associated with *IDH1* mutations [16].

A recent study reported the results of the mutational profiling of 2856 AML cases, including 640 *NPM1*-mut AMLs [17]. The most relevant results of this extensive analysis showed that *NPM1* mutations were (i) significantly co-mutated with *FLT3* and *DNMT3A* mutations; (ii) highly associated with *IDH1* mutations; and (iii) not associated with *RUNX1*, *SRSF2*, *ASXL1,* and *IDH2-R172* mutations [17]. *NPM1* mutations were associated with DNA methylation genes and the activation of signaling genes but exclusively with myeloid transcription factors, spliceosome genes, and chromatin-modifying gene mutations [17].

The favorable prognostic impact of *NPM1* mutations decreases with increasing age of AML patients treated with standard treatments. This finding supported the study of the mutational profile of older *NPM1*-mut AML patients. Thus, one study reported in older *NPM1*-mut AML patients (≥75 years) a significant enrichment of *TET2*, *SRSF2,* and *IDH2* mutations, with a reduced frequency of *DNMT3A* mutations, compared with what was observed in younger *NMP1*-mut AML patients (45% vs. 16%, 22% vs. 3.5%, 28% vs. 12%, and 27% vs. 52%, respectively) [18]. Similar observations were made by Lachowietz et al., who reported a higher frequency of *TET2* and a lower frequency of *DNMT3A* mutations in *NPM1*-mut AML patients ≥65 years compared to those ≤65 years [19]. An extensive analysis carried out on 533 *NPM1*-mutated AML patients showed some notable differences in the mutational profile of patients ≤65 vs. ≥65 years: *TET2* (13% vs. 27%), *NRAS* (13% vs. 7%), *SRSF2* (5% vs. 15%), *WT1* (10% vs. 4%), and *ASXL1* (1% vs. 7%) [20].

Therapy-related AMLs (t-AML) are a heterogeneous group of aggressive myeloid neoplasms occurring in patients with cytotoxic chemotherapy or ionizing radiation. The majority of AMLs display concomitant chromosomal abnormalities and *TP53* alterations; a minority of t-AMLs have a normal karyotype (NK). About 35% of NK t-AMLs have *NPM1* mutations. The mutational spectrum of NK t-AMLs was similar to that observed for NK de novo AMLs, although the frequency of some mutations showed some significant differences: *NPM1* (35% vs. 49%, respectively), *FLT3* (23% vs. 36%, respectively), *KRAS* (12% vs. 5%, respectively), and *GATA2* (9% vs. 2%, respectively) [21]. It was reported that 7–9% of patients developing t-AML display *NPM1* mutations. t-*NPM1*-AMLs are similar to de novo *NPM1*-mut AMLs but different from the rest of t-AMLs: t-*NPM1*-mut AMLs have a normal karyotype more frequently than t-AMLs (88% vs. 28%, respectively); t-*NPM1*-mut AMLs are more frequently associated with *DNMT3A*-mut and *TET2*-mut than t-AMLs (43% vs. 14% and 40% vs. 10%, respectively); t-*NPM1*-mut AMLs are less frequently associated with *TP53* mutations than t-AMLs (3% vs. 35%, respectively) [22].

## 3. Cell Differentiation Heterogeneity of *NPM1*-Mut AMLs

*NPM1-mut* AMLs, in addition to genetic heterogeneity, exhibit a consistent degree of phenotypic heterogeneity, with a subset showing monocytic differentiation and another subset lacking monocytic differentiation and showing a promyelocyte-like CD34^−^/HLA-DR^−^ immunophenotype [2].

Mason et al. explored a possible link between phenotypic and genotypic heterogeneities in a group of 239 *NPM1-mut* AMLs; 41% of these AMLs displayed monocytic differentiation and the remaining 59% of cases were subdivided into two subgroups, one lacking HLA-DR and CD34 expression (double negative (DN), 30% of cases) and the other defined as myeloid (29% of cases) [23]. These three phenotypic subtypes differed for some genotypic features: *TET2* and *IDH1-2* mutations were more frequent in DN cases (96% positivity) than in myeloid (44%) or monocytic (48%) subtypes; *DNMT3A* mutations were significantly less frequent in DN AMLs (27%) than in myeloid (44%) or monocytic cases (54%) [23]. These three phenotypic groups showed also significant differences in their outcome in that the DN-NPM1 displayed a DFS and OS (64.7 and 66.7 months, respectively) longer than monocytic NPM1 (20.6 and 44.3 months, respectively) and myeloid NPM1 (8.4 and 20.2 months, respectively) [23].

Using a machine learning approach for the analysis of gene expression profiles, Mer et al. identified two different subtypes within *NPM1*-mut AML patients, one labeled as primitive and the other one as committed, based on the respective presence or absence of a stem cell signature [24]. *FLT3-ITD* mutations were significantly more frequent in primitive than committed *NPM1* subtypes (62% vs. 28%, respectively), while *DNMT3A* mutations were less common in primitive than in committed *NPM1* subtypes (38% vs. 56%, respectively) [24]. The primitive subtype was associated with a significantly worse survival than the committed subtype; furthermore, the primitive subtype was more sensitive to kinase inhibitors [24].

The analysis of the gene expression profile of *NPM1-mut* AMLs further supported their heterogeneity. Cheng et al. explored the gene expression profiles by RNA sequencing and somatic genomic alterations by targeted or whole-exome sequencing of 655 AML patients and based on enhanced consensus clustering, identified eight stable gene expression subgroups (G1 to G8) [25]. *NPM1-mut* AMLs clustered into three different subgroups (G6, G7, and G8), showing a high expression of *HOXA/B* genes and various differentiation stages, from hematopoietic stem/progenitor cells down to monocytes, and specifically HOX-primitive (G7), HOX-mixed (G8), and HOX-committed (G6); in the G6 and G8 subgroups, *KMT2A* fusions clustered, and in the G7 subgroup, *NUP98* fusions clustered [25]. *NPM1-mut* AMLs present in the G6–G8 subgroups showed relevant differences at the level of their co-mutation profile: (i) *DNMT3A* represented the most frequent co-mutations in the G6 and G8 subgroups, while *NPM1-mut* present in the G7 subgroup rarely associated with *DNMT3A* mutations but frequently associated with *IDH1* and *IDH2* mutations; (ii) the frequency of triple-mutated *NPM1/DNMT3A/FLT3-ITD* was higher in the G8 subgroup (with a percentage of 4.5%, 6%, and 22.2% in G6, G7, and G8, respectively); (iii) the frequency of triple-mutated *NPM1/FLT3-ITD/TET2* or *NPM1/FLT3-ITD/IDH2* was more recurrent in the G7 subgroup compared to the two other subgroups (0%, 28.4%, and 7.6% in G6, G7, and G8, respectively) [25]. Concerning the differentiation state, G6 and G7 exhibited a more differentiated monocytic phenotype and stem cell phenotype, respectively [25]. The comparison of the prognostic profile of these three subgroups showed that patients in G8 (HOX-mixed) have the poorest prognosis, in terms of both OS and EFS, compared to those in G6 (HOX-committed) and G7 (HOX-primitive) [25].

## 4. Clonal Architecture and Clonal Evolution of *NPM1*-Mutant AMLs

The study of *NPM1-mut* AMLs is one of the best models to explore the mechanisms of leukemic clonal evolution. The analysis of the allelic burden (VAF) of *NPM1* mutations and of the associated co-mutations allowed the definition of a mutational clonal hierarchy of *NPM1-mut* at diagnosis. This analysis showed that in the majority of patients, a higher VAF was detected for some co-mutations than for *NPM1*, including *DNMT3A*, *IDH1*, *IDH2*, *SRSF2,* and *TET2*, thus suggesting that mutations in those genes represent first hits and occur at an early phase of leukemic development; on the contrary, other co-mutations such as *FLT3*, *NRAS,* and *WT1* showed a significantly lower VAF than *NPM1-mut*, thus indicating that these are second-hit mutations [26]. According to the VAF, Cappelli et al. distinguished co-mutations occurring in *NPM1-mut* AMLs and distinguished the mutations in CHIP-like, including *DNMT3A*, *TET2*, *ASXL1*, *ID1*, *IDH2*, *SRFSF2,* and *STAG2*, and CHOP-like, including *FLT3*, *GATA2*, *NRAS*, *PTPN11*, *WT1*, *TP53,* and *RUNX1* [27]. The persistence or the acquisition of CHOP-like mutations was associated with an inferior outcome [27].

The study of the clonal architecture of *NPM1-mut* AMLs at a single cell level showed that these leukemias are usually organized following simple clonal architectures with one to six subclones and branching; in all cases studied, *NPM1* mutations were secondary or subclonal to other driver mutations; in a part of these leukemias, it was postulated, through the analysis of single CD34^+^/CCD33^−^ cells, the existence of pre-leukemic cells bearing one or more driver mutations, lacking *NPM1* mutations [28]. Importantly, after transplantation in immunodeficient mice, the dominant regenerative clone in vivo was a *NPM1-mut* subclone, even when *NPM1-mut* was minoritarian at a subclonal level in the diagnostic leukemic cells [28]. According to these findings, a model was proposed in which *NPM1-mut* AMLs develop from pre-existing clonal hematopoiesis [28]. Additional studies supported this clonal evolution model of *NPM1-mut* AMLs. Desai et al. reported the longitudinal history of an AML patient with *IDH2*-mut clonal hematopoiesis who developed AML one year after the acquisition of an *NPM1* mutation [29]. *NPM1-mut* AML patients with concomitant *DNMT3A* mutations, responding optimally to standard induction chemotherapy, despite the persistence of the *DNMT3A* mutations, achieved a long-term response [30,31].

Single-cell mutation analysis provided a fundamental tool to analyze AML clonal evolution. A study by Miles et al., exploring clonal evolution in 123 AML samples, provided evidence that AML development is characterized by a small number of mutant clones, frequently harboring co-occurring mutations in epigenetic regulators [32]. In *NPM1-mut* AMLs with concomitant *FLT3-ITD* mutations, the size of double-mutant *NPM1/FLT3* clones was significantly greater than those of *NPM1* or *FLT3* single-mutant clones; in contrast, in *NPM1-mut* AMLs with concomitant *RAS* mutations, there is evidence of cooperativity between these mutations with respect to single-mutant *RAS* clones, but not to single-mutant *NPM1* clones [32].

The mutational dynamics at the clonal level were explored in *NPM1-mut* AMLs at diagnosis and at relapse. Kronke et al. reported the evaluation of 53 relapsing *NPM1-mut* AMLs and showed that in about 90% of cases the recurrence was related to the original *NPM1-mut* clone at relapse [33]. The relapsed AMLs were characterized by an increased mutational complexity; some mutations, such as *DNMT3A* and *IDH2* mutations, were almost completely stable at relapse, while other mutations, such as *FLT3-ITD* and *RAS* mutations, showed low stability; recurrent genetic alterations acquired at relapse often involved *ETV6*, *TP53*, *NF1*, and *WT1* genes [33]. Larger analyses on relapsing *NPM1-mut* AML patients showed that 9–14% of patients relapsed with *NPM1-WT* AMLs [34,35]. At diagnosis, *FLT3-ITD* mutations were more frequent in patients with *NPM1-mut* at relapse, while *DNMT3A* mutations were more frequent in those relapsing with *NPM1-WT* AML [34]. According to the results of exome sequencing studies, it was proposed that in *NPM1-mut*-persistent patients, an *NPM1-mut* clone survived chemotherapy, showed additional mutational evolution, and subsequently acquired a growth advantage, causing relapse; in *NPM1*-*mut* patients relapsing with *NPM1-WT* AML, the initial *NPM1-mut* clone is eradicated by chemotherapy and the relapse is ensured by a surviving clone with preleukemic mutations acquiring new mutations and leukemic properties [34,35].

## 5. Prognostic Heterogeneity of *NPM1*-Mut AMLs

The analysis of the overall survival of *NPM1*-mut AML patients showed a marked heterogeneity, with a part of patients showing a good OS, with an observed risk hazard distributed between favorable and intermediate ELN risk groups and another part of patients showing a poor OS with an observed risk hazard distributed between intermediate and adverse ELN risk groups [2].

The European HARMONY Alliance retrospectively analyzed a large cohort of 1011 *NPM1*-mut patients for their mutational profile and their response to standard therapy, showing that (i) the triple mutation group *NPM1/DNMT3A/FLT3-ITD^high^* identified a subgroup with adverse prognosis (2-year OS of 25%, similar to that observed for *NPM1/TP53* double-mutant AMLs); (ii) the double mutation groups *FLT3-ITD^low^/DNMT3A* or *FLT3-ITD*^high^/*DNMT3A*-WT exhibited an intermediate prognosis (2-year OS of 45% and 53%, respectively); (iii) *NRAS*, *KRAS*, *PTPN11* or *RAD21* mutations were associated with a better OS (however, these mutations did not affect prognosis in the presence of the triple mutation group *NPM1/DNMT3A/FLT3-ITD*) [36]. Using this large database, a machine learning algorithm was developed, allowing the identification of combinations of up to four co-mutations with prognostic significance [37]. This algorithm allowed the stratification of *NPM1-mut* AML patients into four groups with increasing prognostic adversity: favorable, intermediate-1, intermediate-2, and adverse (Figure 2). In particular, the triple combination *NPM1-mut/FLT3-ITD/DNMT3A-mut* identified a subgroup with adverse prognosis (2-year OS of 33%), similar to that observed for the small subgroup (1.5% of total *NPM1-mut* AMLs) (Figure 2). Two subgroups were identified in the favorable group: a first subgroup, involving *TP53-WT/FLT3-WT/DNMT3A-mut* and *NRAS* or *KRAS* or *PTPN11* or *RAD21*-mutated AMLs, and a second subgroup, involving *TP53-WT/DNMT3A-WT/IDH-mut* patients. The intermediate-1 group involves two subgroups: one composed of AMLs with a *TP53-WT/FLT3-WTR/DNMT3A-WT* mutational profile; the other one involves *TP53-WT/FLçT3-WT/DNMT3A-mut/IDH-mut* and *NRAS/KRAS/PTPN11/RAD21-WT* cases (Figure 2). The intermediate-2 group involves two subgroups: one subgroup implies AMLs *TP53-WT/FLT3-WT/DNMT3A-mut/IDH-mut*; the other subgroup is composed of AMLs *TP53-WT/FLT3-ITD/DNMT3A-WT/IDH-WT* (Figure 2). The 3-year OS of these groups was 78%, 63%, 48%, and 29% for favorable, intermediate-1, intermediate-2, and adverse groups, respectively [37]. The prognostic predictive capacity of this algorithm was evaluated in other datasets of *NPM1-mut* AML patients with available genetic and clinical information [37].

Mrozek et al. reported in 1637 adult AML patients the evaluation of the 2022 ELN stratification risk system [38]. *NPM1-mut/FLT3-ITD*-negative was included in the favorable group; the outcome of these patients harboring myelodysplasia-related mutations was worse than the outcome of the patients without myelodysplasia-related mutations (CR rates: 67% vs. 81%, respectively; PFS: 30% vs. 43%, respectively; 5-year OS: 32% vs. 42%, respectively) [38]. The outcome of the patients was similar to that of patients classified as intermediate-risk patients following ELN 2022 [38]. According to the ELN 2022 guidelines for AML, *NPM1-mut* AML patients with adverse cytogenetics are classified as adverse-risk patients [5]. The evaluation of 13 of these patients showed that they have a shorter 5-year OS and DFS compared to that observed in *NPM1-mut* patients without adverse risk cytogenetics (23% vs. 41% and 38% vs. 41%, respectively) [38]. A comparison with other AML groups with intermediate or adverse risk shows that these *NPM1-mut* AML patients with adverse cytogenetics are more similar to patients with intermediate risk than to those with adverse risk [38].

Angenendt et al. re-evaluated the data on chromosomal abnormalities in 2426 patients with *NPM1-mut* AMLs, upgrading the risk category based on chromosomal abnormalities evaluated following ELN 2022 [8,39]. In these patients, adverse cytogenetics according to ELN 2022 were associated with lower complete remission rates (87%, 85%, and 66% for normal, aberrant intermediate, and adverse karyotypes, respectively) and inferior overall survival (40%, 36%, and 16%, for normal, intermediate, and adverse karyotypes, respectively) [39].

Several studies have explored a possible prognostic impact of *NPM1-mut* VAF, generating conflicting results. *NPM1-mut* VAF was shown to positively correlate with leukemic cellularity at diagnosis and the percentage of leukemic blasts in peripheral blood and to negatively correlate with platelet counts [40,41,42]. Patel et al. explored 109 patients with de novo *NPM1-mut* VAF (≥0.44) correlated with shortened OS and EFS compared to the rest of *NPM1-mut* AMLs; high *NPM1-mut* VAF had a particularly negative prognostic impact in *NPM1-mut* patients treated with stem cell transplantation in first remission and in patients with mutated *DNMT3A* [40]. In a second study, the same authors showed that high *NPM1-mut* VAF correlated with minimal residual disease (MRD) at first remission; both *NPM1-mut* VAF and MRD at first remission predicted a shortened EFS [43].

Abbas et al. reached a different conclusion in their evaluation of 147 *NPM1-mut* AML patients treated with induction chemotherapy. First, they observed a significantly higher *NPM1-mut* VAF in patients with *FLT3-ITD* compared to those with *FLT3-WT* (42.7% vs. 39.1%, respectively); however, *NPM1-mut* VAF did not correlate with the *DNMT3A* mutational status or with the presence of cytogenetic abnormalities [41]. No significant correlation was observed between the level of *NPM1-mut* VAF and either OS or EFS in the entire cohort of patients or in any subgroup [40]. Discrepancies between this study and the previous study could be related to differences in induction chemotherapy regimens used in these two studies [41].

Rothenberg-Thurley et al. reported the study of 417 *NPM1-mut* patients, and the analysis of their *NPM1-mut* VAF showed that the median *NPM1-mut* VAF was 0.43 and was higher in type A than in type B *NPM1* mutations and was not associated with an abnormal karyotype; patients with high *NPM1-mut* VAF more frequently had concomitant *FLT3-ITD* (47% vs. 37%) and *DNMT3A* (63% vs. 46%) mutations compared to those of patients with low *NPM1-mut* VAF; a high *NPM1-mut* VAF associated with shorter OS [42]. However, in multivariate analysis, after adjusting for the *FLT3-ITD* allelic ratio and/or *DNMT3A* mutational status, only these genetic alterations and not *NPM1-mut* VAF remained associated with OS [42]. According to these results, it was suggested that high *NPM1-mut* VAF may simply represent a marker of highly proliferative subsets of *NPM1-mut* AMLs, such as those with *FLT3-ITD* mutations, rather than an independent prognostic factor [42].

## 6. *DNMT3A* Mutations in NPM1-Mut AMLs

*DNMT3A* mutations represent the mutations most frequently associated with *NPM1* mutations in AMLs. *DNMT3A* mutations are frequently observed in aging individuals without overt leukemia and in association with clonal hematopoiesis of undetermined potential (CHIP). *DNMT3A* gene mutations precede *NPM1* mutations, exerting a stimulatory effect on the self-renewal of leukemic clones. In the ELN 2022 leukemia classification, *DNMT3A* is not considered a high-risk mutation.

In AML patients, the levels of *DNMT3A* mutations do not correlate with presenting clinical features or concurrent gene mutations and do not affect the OS; *DNMT3A-mut* expression persists in most AML patients achieving complete remission after induction chemotherapy, suggesting the persistence of clonal hematopoiesis in hematological remission [31].

Cappelli et al. retrospectively analyzed a large cohort of 1977 *NPM1-mut* AML patients [44]. In these patients, the *DNMT3A* gene was the most frequently co-mutated (45% of cases). The VAF of the *DNMT3A* mutation was significantly higher than that of *NPM1* mutations, thus indicating that they precede *NPM1* mutations. *DNMT3A* mutations displayed a peculiar pattern according to age, being more frequent in younger than older (≥60 years) patients: 51% vs. 40%, respectively [43]. This pattern of *DNMT3A* mutational frequency was dependent on the type of mutations: in fact, *DNMT3A-R882* mutations were more frequent in younger than older *NPM1-mut* patients (58% vs. 42%, respectively), while non-*R882 DNMT3A* mutations were less frequent in young than in older patients (39% vs. 61%, respectively) [44]. In contrast, other CHIP-related genes, such as *ASXL1* and *TET2,* were less frequently mutated in younger than in older *NPM1-mut* AML patients (1% vs. 4% and 17% vs. 27%, respectively) [44]. Importantly, in *NPM1-WT* AMLs, the frequency of *DNMT3A* mutations increased with age [44]. In addition to *DNMT3A*, other gene mutations were preferentially associated with younger age, such as *WT1* (8% vs. 3%), *NRAS* (25% vs. 17%), and *PTPN11* (8% vs. 1%) [44]. The co-mutational pattern of *NPM1-mut/DNMT3A-mut* double-mutated AMLs was significantly different compared to *NPM1-mut/DNMT3A-WT*; in fact, *NPM1-mut/DNMT3A-mut* was positively associated with *FLT3-ITD*, *NRAS,* and *PTPN11* mutations and negatively associated with *IDH2R140*, *STAG2,* and *SRFSF2* mutations [44]. The analysis of survival according to the *DNMT3A* mutational status showed that in *NPM1-mut* patients, *DNMT3A* mutations were not associated with survival irrespective of the *DNMT3A* mutational subtype; the presence of *FLT3-ITD* mutations had a detrimental effect on both *DNMT3A-mut* and *DNMT3A-WT NPM1-mut* patients; particularly, the presence of the *DNMT3A-R882* mutation in association with *FLT3-ITD* was associated with worse outcomes [44].

In a more recent study, the same authors reported that 150 *NPM1-mut* AML patients achieved CR following induction chemotherapy: patients with CHIP mutations, such as *DNMT3A*, *TET2*, *ASXL1*, *IDH1*, *IDH2,* and *SRSF2,* had a frequency of relapse and a probability of OS comparable to that observed for *NPM1-mut* without co-mutations at remission; in contrast, patients with mutations that were not CHIP related, such as *FLT3-ITD*, *FLT3-TKD*, *GATA2*, *NRAS*, *PTPN11*, *WT1*, *TP53,* and *RUNX1*, persistent at remission or acquired at relapse, had an increased probability of relapse and a poor prognosis [27]. These non-CHIP mutations were defined as CHOP mutations. Finally, this study showed that the persistence of *DNMT3A-R882* mutations was not associated with inferior survival [27]. These observations have important implications for monitoring *NPM1-mut* AMLs during treatment [45].

Onate and coworkers explored the prognostic impact of *DMNT3A* mutations in *NPM1-mut* AMLs subdivided into three subgroups according to the FLT3 mutational status: *DNMT3A-FLT3-WT*, *DNMT3A-FLT3-ITD^low^*, *DNMT3A-FLT3-ITD^high^;* patients with *DNMT3A* mutation had a delayed *NPM1-mut* clearance after induction chemotherapy, but *DNMT3A* mutations did not modify the prognostic value of the *FLT3-ITD* allelic ratio in *NPM1-mut* AMLs [46].

The characterization of *DNMT3A-mut/NPM1-mut* AMLs has led to the identification of an AML subset characterized by triple positivity for *NPM1*, *DNMT3A,* and *FLT3-ITD* mutations. A part of these triple-positive AMLs also displays either *TET2* or *WT1* mutations. An initial study by Loghavi and coworkers suggested that triple-positive *NPM1/DNMT3A/FLT3-ITD* may represent a peculiar subset of *NPM1-mut* AMLs associated with poor prognosis: in fact, these AMLs displayed an OS shorter than that observed in double-positive *NPM1-mut/FLT3-ITD* AMLs [47]. The concomitant presence of *NPM1/DNMT3A/FLT3-ITD* mutations was observed in about 6% of AMLs, characterized by a high frequency of leukemia stem cells, an aberrant immunophenotype (with low CD34 expression, associated with high CD56 expression), and a high expression of hepatic leukemia factor (whose expression is required for the maintenance and the expansion of leukemic stem cells) [48].

Several studies have reported the poor overall survival of triple-mutant *NPM1/DNMT3A/FLT3-ITD* patients: Bezerra et al. reported a 5-year OS of only 4% for these patients, an increased risk of relapse, and a lower disease-free survival [49]. In this study, the analysis was limited to AML patients bearing *R882-DNMT3A* mutations, the only mutations of *DNMT3A* having biochemical changes [50] and consequences for clonal hematopoiesis [51].

Wakita and coworkers retrospectively analyzed 605 Japanese patients with de novo *AML* (*174* with NPM1-mut AML) [52]. The analysis of both *NPM1-mut* and *NPM1-WT* AML patients showed that the presence of *DNMT3A-R882* mutations was associated with a reduced overall survival compared to the respective *DNMT3A-WT* patients; in both *NPM1-mut/DNMT3A-WT* and *NPM1-mut/DNMT3A-R882* AMLs, the co-occurrence of *FLT3-ITD* mutations, at both low and high allelic ratios, significantly reduced OS; triple-mutant *NPM1/DNMT3A/FLT3-ITD* patients showed a marked decline in OS [52].

## 7. *FLT3* Mutations in NPM1-mut AML Patients

Two types of *FLT3* mutations are observed in AMLs: internal tandem duplication of the juxta membrane domain (*FLT3-ITD*) and point mutations or the deletion of the tyrosine kinase domain (*FLT3-TKD*)*. FLT3* mutations are very frequent in *NPM1-mut* patients: *FLT3-ITD* (41%), *FLT3-TKD* (21%), and *FLT3-ITD/FLT3-TKD* (4.5%) [15]. *FLT3-ITD* mutations may occur at the level of the juxtamembrane domain (*FLT3-ITD-JMD*) or at the level of tyrosine kinase domain 1 (*FLT3-ITD-TKD1*) or in both of these regions of *FLT3* (*FLT3-ITD-JMD-TKD1*). In the RATIFY trial enrolling a large cohort of *FLT3-ITD*-mutated patients, it was reported that in *NPM1-mut/FLT3-ITD* patients, 60.5% displayed *FLT3-ITD-JMD* mutations, 17% displayed *FLT3-ITD-TKD1,* and 22.5% displayed *FLT3-ITD-JMD-TKD1* [53]. *FLT3-ITD-TKD1* co-mutations were associated with a worse prognosis.

Patients with *NPM1-mut/FLT3-ITD* and *NPM1-mut/FLT3-TKD* have a similar overall survival when treated with intensive frontline therapy, while patients displaying concomitant *FLT3-ITD* and *FLT3-TKD* mutations have a dismal overall survival [54].

The ELN 2017 classification supported the evaluation of the *FLT3-ITD* allelic ratio as a prognostic parameter, classifying patients with a high *FLT3-ITD* ratio in a worse category group. *FLT3-ITD^high^* was associated with a higher WBC, higher blood and bone marrow blasts, and with more frequent *NPM1* mutations, while *FLT3-ITD^low^* was associated with *FLT3-TKD* [55]. However, in spite these clinico-biological differences, the outcomes of AML patients undergoing allogeneic HSC were similar for both *FLT3^low^* and *FLT3^high^* AML patients [55].

The evaluation of *FLT3-ITD* minimal residual disease by NGS in complete remission represents the best and most sensitive biomarker to predict the outcomes of these patients [56].

## 8. *IDH1* and *IDH2* Mutations in *NPM1*-Mut AMLs

About 25% of *NPM1*-mut AMLs have a mutation of the *IDH1* or *IDH2* gene. *IDH1* and *IDH2* mutations also occur in a part of cases in association with *DNMT3A* mutations while in other patients are co-mutated with *FLT3-ITD* or *FLT3-TKD* [15]. In *NPM1*-mut AMLs, the most frequent *IDH1* mutations are represented by *IDH1R132H*, while those represented by *IDH1R132C* are less frequent; the most frequent *IDH2* mutations are represented by *IDH2R140Q*, while *IDH2R172K* mutations are only rarely observed in *NPM1*-mut AMLs [56]. Importantly, the association of *IDH1/IDH2* mutations with *NPM1* mutations improved their prognostic impact in comparison with the prognosis of AMLs with the same type of *IDH1* or *IDH2* mutation but in association with other co-mutations [57].

*IDH1* mutations occur in about 7–8% of AML patients, mostly associated with a normal karyotype. *NPM1* and *DNMT3A* gene mutations are most frequently associated with *IDH1* mutations. In particular, 66% of *IDH1-mut* AMLs displayed *NPM1-mut; IDH1-R132H* was strongly associated with *NPM1-mut* (89% of cases), while *IDH1-R132H* was associated with *NPM1-mut* in 28.5% of cases; other rarer *IDH1* mutations were also strongly associated with *NPM1-mut* (75% of cases) [58]. *IDH2R140* mutations were associated with *NPM1* mutations in about 50% of cases, and in *NPM1-mut,* AMLs were associated with frequent *DNMT3A*, *FLT3-ITD,* and *SRSF2* co-mutations [59]. These findings indicate that the association between *IDH1/IDH2* and *NPM1* mutations is stronger than the association between *NPM1* and *IDH1/IDH2* mutations.

Mason et al. distinguished two subtypes of *NPM1-mut* AMLs according to their immunophenotypic features: the acute promyelocytic-like subtype, characterized by the absence of CD34 and HLA-DR expression and strong myeloperoxidase expression, was highly enriched in *IDH1*, *IDH2* or *TET2* co-mutated cases [23]. This APL-like subtype is associated with longer relapse-free and overall survival compared with cases that were positive for CD34 and/or HLA-DR [23].

In conclusion, *NPM1-mut* AMLs with *IDH1* or *IDH2* co-mutations do not seem to have a worse prognosis compared to *NPM1-mut* AMLs without *IDH1* or *IDH2* mutations.

## 9. Cohesin Complex Gene Mutations in NPM1-mut AMLs

Cohesin complex genes, *STAG2*, *RAD21*, *SMC1A,* and *SMC3* mutations, are observed in about 19% of *NPM1-mut* AMLs [15]. Complex cohesin genes are mutated in about 11% of all AMLs [59]. Some of the cohesin genes, including *RAD21*, *SMC1A,* and *SMC3,* display the highest frequency of mutations in *NPM1-mut* AMLs [60].

*STAG2* mutations occur in about 3% of *NPM1-mut* AMLs [15]; *NPM1-mut* AMLs represent 15% of all *STAG2-mut* AMLs [59]. *NPM1* is less commonly mutated in *STAG2-mut* AMLs than in the rest of AMLs (15% vs. 32%, respectively). *NPM1-mut* AMLs with *STAG2* mutations also frequently display *FLT3-ITD* and *NRAS* mutations and, more rarely, *DNMT3A* mutations.

*RAD21* mutations occur in about 6% of *NPM1*-mut AMLs [15]; *NPM1-mut* AMLs represent 57% of all *RAD21-mut* AMLs [59]. *NPM1* is significantly more frequently mutated in *RAD21-mut* AMLs than in the rest of AMLs (57% vs. 30%, respectively) [60]. Double-mutant *NPM1-RAD21* AMLs display a pattern of associated mutations comparable to that observed in the whole group of *NPM1-mut* AMLs [61].

*SMC3* is mutated in 4.5% of *NPM1-mut* AMLs [60].; *NPM1-mut* AMLs represent 65% of all *SMC3-mut* AMLs [59]. Double-mutant *NPM1/SMC3* frequently display additional mutations of *NRAS* [59].

*SMC1A* is mutated in about 5% of *NPM1-mut* AMLs [15]; *NPM1-mut* AMLs represent 40% of all *SMC1A-mut* AMLs [60].

Simonetti et al. explored the metabolomic profile of AMLs and, through an integrated analysis of genomic–metabolic profiles defined two subgroups of *NPM1-mut* AMLs, one of these two subgroups was enriched in cohesin/DNA damage-related genes and showed a higher mutation load, transcriptomic signatures of a reduced inflammatory state, and better ex vivo response to EGFR and MET inhibition [61].

Benard et al. explored the genomic data of 2829 AML patients and reached the conclusion that clonal architecture represents a predictive parameter of clinical outcomes and drug sensitivity [62]. In some instances, the order of mutations in functional classes stratified survival: this was the case of patients with co-occurring mutations in *NPM1* and chromatin/cohesin complex genes; in these patients, if a chromatin/cohesin mutation occurred before an *NPM1* variant, there was a strong association with poor survival [62].

Studies in inducible mouse models of *NPM1-mut/SMC3-mut* have shown that cohesin gene mutations alter the transcriptome in the context of the *NPM1*-mutant; in particular, it was shown that the Rac 1-2 exchange factor Dock1 is specifically upregulated in double-mutant *NPM1/SMC3* cells and could represent a therapeutic target in these leukemias [63].

## 10. RAS Mutations in NPM1-Mut AMLs

*RAS* genes are frequently mutated in *NPM1-mut* AMLs: *NRAS* in about 20% of cases and *KRAS* in about 4% of cases [15]. *NRAS* mutations in these patients are frequently associated with *DNMT3A* and *PTPN11* mutations; *NRAS* is rarely co-mutated with *FLT3-ITD* or *FLT3-TKD*. *NPM1* mutations preferentially associate with *NRAS^G12/13^* mutations but not with *NRAS^Q21^* mutations [1]. *NRAS* mutations do not affect the outcomes of *NPM1-mut* AMLs; *NPM1/DNMT3A/NRAS* triple-mutant AMLs are associated with a favorable prognosis [64].

Rivera et al. explored 273 de novo AML patients treated with induction therapy and showed that in these patients, a favorable karyotype and concomitant *NPM1* and *NRAS* mutations were associated with higher CR, ORR, and OS [65].

## 11. Myelodysplasia-Related Alterations in NPM1-Mut AMLs

AML with myelodysplastic changes (AML-MRC) is a subgroup of AMLs usually associated with poor prognosis. The diagnosis of AML-MRC encompasses a variety of AMLs based on three main criteria: a history of a myelodysplastic syndrome or of myelodysplastic/myeloproliferative neoplasm (AML-MRC-H); the presence of MDS-defining cytogenetic abnormality (AML-MRC-C); the morphological detection of multilineage dysplasia (AML-MRC-M) [66].

According to these criteria, four different AML-MRC subtypes can be identified: AML-MRC-C, AML-MRC-H, AML-MRC-M, and AML-MRC-TS (this last subtype identifies AMLs originated from previously treated MDS or MDS-MPN).

### 11.1. NPM1-Mut MDS

In contrast to AMLs in which *NPM1* mutations are frequent, myelodysplastic syndromes (MDS) only rarely display *NPM1* mutations. In fact, the frequency of *NPM1-mut* in patients with a diagnosis of MDS or myelodysplastic/myeloproliferative neoplasm (MDS/MPN) is low, ranging from 0% to 9% [67,68,69]. *NPM1-mut* MDS or MDS/MPN exhibit an aggressive clinical course with a high rate of transformation to AML [69]. Forghieri et al. proposed that *NPM1-mut* MDS or MDS/MPN may be classified as AML, even in the presence of <20% bone marrow blasts [69]. Since the NPM1-mut AML needs <20% blasts in the bone marrow for diagnosis, the entity of MDS with *NPM1-mut* becomes challenging.

Maurya et al. explored 111 MDS patients and reported an *NPM1-mut* frequency of 3.6%, with 50% of *NPM1-mut* patients showing an IPSS low risk and 50% an IPSS high risk and a reduced OS compared to MDS patients without *NPM1* mutations [70].

Montalban-Bravo et al. reported the analysis of 31 *NPM1-mut* MDS patients observed in a cohort of 1900 MDS patients [71]. This analysis included the largest series of *NPM1-mut* MDS patients reported thus far. These patients were predominantly classified as intermediate risk and high risk, with a median BM blast percentage of 10% and with a normal karyotype in 77% of cases; compared to the rest of the MDS patients, *NPM1-mut* MDS patients were younger, had lower hemoglobin levels, had a higher median BM blast percentage at diagnosis, and had a higher frequency of a normal karyotype [71]. In addition, 38% of these patients had a transformation to AML after a median of 14 months, maintaining in all cases the *NPM1* mutation [70]. The analysis of the mutational profile of these patients showed a pattern of mutations similar to that observed in *NPM1-mut* AMLs: in fact, frequent *NRAS* (32%), *DNMT3A* (27%), *TET2* (18%), *WT1* (18%), *PTPN11* (*13%*), *FLT3* (13%), and *IDH2* (10%) mutations were observed [71]. Further, 32% of these patients received cytotoxic chemotherapy and 65% hypomethylating agents: patients treated with chemotherapy had higher complete response rates and longer overall survival compared to those treated with hypomethylating agents (90% vs. 28% and not reached vs. 16 months, respectively) [71]. These observations, although based on a retrospective analysis, strongly support the treatment of *NPM1-mut* MDS patients with intensive chemotherapy, possibly followed by allogeneic SCT [71].

Another recent study showed a very low response rate of *NPM1-mut* MDS patients to hypomethylating agents, while *NPM1-mut* sAMLs treated with intensive chemotherapy showed a high rate of complete responses with 39% of patients undergoing allogeneic SCT [72]. These findings, together with previous studies, support the view that *NPM1-mut* MDSs are an aggressive clinicopathologic entity requiring, if clinically suitable, treatment with intensive chemotherapy [72].

A second study reported the characterization at clinical and molecular levels of a consistent number (45) of *NPM1-mut* MDS. *NPM1-mut* MDS compared to *NPM1-WT* MDS were associated with younger age, lower WBC, and bone marrow cellularity at diagnosis [73]. NGS studies showed some remarkable differences between *NPM1-mut* and *NPM1-WT* MDS: *IDH1*, *IDH2*, *ASXL1*, *RUNX1,* and *TP53* mutations were less frequent in *NPM1-mut* MDS than in *NPM1-WT* MDS; *PTPN11* and *DNMT3A* mutations were more frequent in *NPM1-mut* than in *NPM1-WT* MDS [73] (Figure 3). The frequency of patients with an abnormal karyotype was markedly lower in *NPM1-mut* than in *NPM1-WT* MDS (12% vs. 61%, respectively) [73]. For the mutational profile and for chromosomic abnormalities, *NPM1-mut* MDS were more similar to *NPM1-mut* AML than to *NPM1-WT* MDS [73].

Based on the findings of these studies, Falini et al. proposed that *NPM1-mut* MDS represent *NPM1-mut* AML diagnosed at an early stage and that must be treated with intensive chemotherapy, followed by allogeneic SCT, as typical *NPM1-mut* AMLs [74].

### 11.2. NPM1-Mut AMLs with MDS-Related Mutations

AMLs with MDS-related gene mutations represent a heterogeneous group of AMLs and can be collectively defined as sAML-like. These AMLs are characterized by the occurrence of one or more mutations of MDS-related genes, including *SRFSF2*, *SF3B1*, *U2AF1*, *ZRSR2*, *ASXL1*, *EZH2*, *BCOR,* and *STAG2*, defined by Lindsley et al. to be more than 95% specific for AMLs evolved from MDS or MDS/MPN [75]. The AMLs bearing mutations of MDS-related genes correspond to different molecular groups, including *NPM1-mut* AMLs.

Although we cannot exclude the possibility of the presence of an unrecognized antecedent MDS before AML diagnosis in patients with sAML mutations, these mutations can be detected in more than 30% of patients rigorously defined as de novo AMLs [75,76,77]. As shown by Gardin et al. [76] and by Fuhrmam et al. [76], sAML-like encompasses most AMLs defined as AML-MRC-H and AML-MRC-M; however, only a part of AML-MRC-C display MDS-related mutations [77].

Recently, AMLs harboring eight sAML mutations and *RUNX1* mutation have been categorized as AML with MDS-related gene mutations in the International Consensus Classification (ICC) [3] and the adverse-risk group in the 2022 ELN risk classification [5]. It is important to note that in these new classifications of AMLs, the definition of AML-MRC abandoned the morphological criteria of dysplasia and adopted the molecular genetics criteria based on the mutational profile. Because of these changes, AML-MRC is now included into genetically defined AML.

A recent study carried out on 1213 Chinese patients with de novo AML reported a frequency of AMLs with MDS-related gene mutations of 3.1% in younger *NPM1-mut* AML patients and of 13.5% in older *NPM1-mut* AML patients; these MDS-related gene mutations were significantly less frequent in *NPM1-mut* AML patients than in *NPM1-WT* AML patients [78].

MDS-related gene mutations are considered a negative prognostic factor. Chan et al. reported the analysis of the clinical response in 233 *NPM1-mut* AML patients whose mutational profile was characterized by NGS; at the mutational level, AMLs with sMut differed from those without sMut for a lower frequency of *DNMT3A* and *FLT3-ITD* mutations (35% vs. 52% and 25% vs. 36%, respectively) and for a higher frequency of *TET2* mutations (39% vs. 23%, respectively); patients with sMut (18.5% of total) achieved a CR of about 70% compared to 85% of patients without sMut; 35% of patients with sMut proceeded to allo-SCT compared to 50% of patients without sMut; the OS of patients with sMut was significantly shorter than that of patients without sMut (12.3 months vs. 73.9 months) [79]. Multivariate analysis showed a negative impact of sMut on overall survival [79]. In the groups of patients with or without sMut, MRD negativity predicted longer OS compared to those with MRD positivity: in the sMut subset, the OS of patients with MRD negativity was 73.9 months compared to 12.3 months for patients with MRD positivity [79].

Wang et al. evaluated a cohort of 107 *NPM1-mut* AMLs characterized for the mutational profile by NGS and treated with risk-adapted therapy; the patients were subdivided into three groups: patients without MDS-related gene mutations (group A), patients with concurrent *FLT3-ITD* mutations (group B), and patients with MDS-related mutations (group C) [80]. Group C patients showed a significant reduction of EFS and an almost significant decrease in OS compared to patients of group A; the therapeutic response of group B and C patients was similar [80].

In conclusion, these studies suggest that MDS-related gene mutations are associated with an inferior survival in *NPM1-mut* AML.

### 11.3. NPM1-Mut Secondary AMLs

It is unclear if *NPM1* mutations remain a positive prognostic indicator within secondary AMLs evolving from a prior myeloid neoplasm. Smith et al. retrospectively analyzed a group of 54 *NPM1-mut* sAMLs evolving from MDS, CMML, MPN, and atypical CML [81]. Compared to de novo *NPM1-mut* AMLs, *NPM1-mut* sAMLs are similar concerning their classification according to ELN 2017, karyotype abnormalities, and the occurrence of *FLT3-ITD* mutations; however, *NPM1mut* sAMLs, compared to *NPM1-mut* de novo AMLs, are more likely to have *RUNX1* mutations (14% vs. 3%, respectively), *CBL* mutations (95% vs. 2%), and *NRAS* mutations (26% vs. 11%) [81]. The overall survival of *NPM1-mut* sAMLs was similar to that observed for *NPM1-mut* de novo AMLs (2-year survival: 46% for sAML and 56% for de novo AML) [81].

Another recent study provided preliminary evidence that *NPM1-mut* sAMLs are molecularly heterogeneous, with only a part of patients bearing sMut: patients with sMut exhibited a significant lower OS compared to those without sMut [82]. This property is not unique to *NPM1-mut* sAMLs but is also observed in *NPM1-mut* de novo AMLs [82].

## 12. TET2 Mutations in IDH1-Mut AML

*TET2* mutations are observed in about 16% of de novo AMLs and are usually associated with clinical signs of hyperleukocytosis, a high blast percentage, and a normal karyotype and are mutually exclusive with *IDH* mutations [1]. In *Npm1-mut* AMLs, about 15% of cases display *TET2* mutations [15]. A significant proportion of *NPM1-mut/TET2-mut* AMLs display additional co-mutations at the level of *DNMT3A:* in both *DNMT3A-mut* and *DNMT3A-WT* cases, *FLT3-ITD* and *FLT3-TKD* are also frequent co-mutational events.

Studies of the molecular characterization of AML patients with hyperleukocytosis at diagnosis have shown the frequent involvement of AMLs with a normal karyotype [83,84]. In particular, 75%, 73%, and 45% of patients with a normal karyotype and hyperleukocytosis at diagnosis were *NPM1-mut*, *FLT3-mut* and *TET2-mut*, respectively; 25% of these patients had concomitant *NPM1* and *TET2* mutations and 21% concomitant *NPM1*, *TET2,* and *FLT3* mutations [84]. A recent study showed that the combined mutation of *NPM1/TET2/FLT3-ITD* and *DNMT3A* observed in about 4% of *NPM1-mut* AMLs resulted in an aggressive leukemia phenotype [85].

In *NPM1-mut* AML patients, the presence of *TET2* mutations was associated with unfavorable prognosis, particularly when these mutations were also associated with *FLT3-ITD* mutations [86,87].

## 13. PTPN11 Mutations in NPM1-Mut AMLs

*PTPN11* mutations are observed in about 7–10% of adult AML patients [88,89]. *PTPN11* mutations are observed in about 17% of *NPM1-mut* AMNLs [15]; in *PTP11-mut* AMLs, *NPM1-mut* AMLs are very frequent (60–65% of cases) and represent the most frequent co-mutations, followed by *DNMT3A*, *NRAS*, *FLT3-ITD*, *IDH2,* and *TET2* mutations [87,88]. *PTPN11-mut* AMLs can be subdivided into two subgroups according to the presence of *NPM1* mutations: *DNMT3A* and *FLT3-ITD* mutations are more frequent in the *PTPN11-mut/NPM1-mut* subgroup than in the *PTPN11-mut/NPM1-WT* subgroup, while the contrary has been reported for *BCOR*, *RUNX1*, *ASXL1* and *SF3B1* mutations [88,89].

*PTPN11mut* AMLs are most frequent in the AMLs classified in the favorable risk genetic group following the ELN risk classification and are associated with higher leukocyte counts [88,89].

The clinical impact of *PTPN11* mutations in AML patients was recently explored. Most of these studies provided evidence that *PTPN11* mutations had an adverse effect on overall survival and a negative prognostic effect on event-free survival [88,89,90,91]. However, Metzeler et al. failed to confirm these results and observed no negative clinical impact of *PTPN11* mutations in a cohort of 116 newly diagnosed *PTPN11-mut* AML patients [92]. Finally, Fobare et al., in their analysis on 1725 newly diagnosed AML patients, including 140 *PTPN11-mut* AML patients treated with intensive induction chemotherapy, showed that *PTPN11* mutations did not affect the outcomes of *NPM1-mut* patients, but had an adverse effect on *NPM1-WT* patients [89]. This differential sensitivity of *PTPN11-mut/NPM1-mut* vs. *PTPN11-mut/NPM1-WT* patients seems to be related to the selective enrichment in *PTPN11/NPM1-WT* patients of co-mutations, such as *BCOR*, *RUNX1*, and *TP53*, associated with adverse outcomes [89].

## 14. Therapy of NPM1-Mut AMLs

The standard therapy for *NPM1-mut* AML patients includes “3 + 7”-induction chemotherapy and consolidation therapy. It was estimated in these patients, there was a complete remission rate of about 80% and an overall survival rate of about 40%. However, more than 50% of *NPM1-mut* AML patients relapse; thus, for high-risk *NPM1-mut* patients, allogeneic stem cell transplantation (allo-HASCT) and additional treatments (such as FLT3 inhibitors) are important therapeutic choices.

In a cohort of 1570 AML patients, *NPM1-mut* cases displayed a favorable prognosis, with a hazard ratio of death of 0.7 and a median OS of nearly 6 years compared to about 2 years in those with *NPM1-WT* AML [1]. The prognostic impact of *NPM1* VAF of *NPM1-mut* AMLs is unclear and should be not used to stratify the risk status of *NPM1-mut* AML patients. This favorable prognostic index was much more pronounced in younger than in older patients: Mrozek et al. showed an OS of 10.5 years in patients with a median age of 44 years compared to 1.7 years in patients with a median age of 69 years [93].

Recent clinical studies have shown the efficacy of venetoclax (an inhibitor of the anti-apoptotic Bcl-2 protein) in the treatment of older *NPM1-mut* AML patients when administered together with hypomethylating agents [94,95] or low-dose AraC [96,97] or intensive chemotherapy [98,99]. An updated analysis of the CAVEAT trial implying the treatment of elderly AML patients treated with intensive chemotherapy and ≥12 months of VEN-based therapy showed that 45% of patients responding to therapy ceased treatment: >50% of these patients remained in remission after ceasing treatment (treatment-free remission, TFR) [100]. Most patients with TFR displayed *NPM1* and/or *IDH2* mutations at diagnosis [100]. A retrospective analysis compared outcomes of *NPM1-mut* AML patients treated with three different regimens (intensive chemotherapy, hypomethylating agents alone, and venetoclax plus hypomethylating agents): venetoclax plus hypomethylating agents improved OS compared to hypomethylating agents alone or to intensive chemotherapy [100]. In particular, in patients treated with Ven + HMA, an OS of 80% after median 1-year follow-up was observed [19,101].

A recent study retrospectively explored the response to venetoclax-based regimens in a group of 206 relapsing/refractory *NPM1-mut* AML patients in comparison with a group of *NPM1-WT AML* patients: high-intensity but not low-intensity regimens were associated with a higher rate of complete responses in *NPM1-mut* patients compared to *NPM1-WT* patients (63% vs. 37%, respectively, for high-intensity regimens); the addition of venetoclax to low-intensity regimens significantly improved the CR and OS in *NPM1-mut* but not in *NPM1-WT* patients (71% vs. 32%, respectively, for CR; 14.7 months vs. 5.9 months, respectively for OS) [102].

High CD33 expression in *NPM1-mut* AMLs provided a rationale supporting the evaluation of the drug-conjugated anti-CD33 antibody gentuzumab ozogamicin (GO) in this AML subtype. The prospective randomized AMLSG 09-09 phase >III study evaluated the efficacy of induction therapy with idarubicin, cytarabine, and all-trans retinoic acid with or without GO; the early death rate was higher in the GO arm compared with the standard arm, while the incidence of relapse in patients achieving complete remission was lower in the GO arm compared to the standard arm [103]. GO failed to improve the event-free survival (EFS) rate; subgroup analysis showed an improvement of EFS in *FLT3-ITD*-negative patients induced by GO [103]. The analysis of MRD levels showed that GO addition reduced the residual *NPM1-mut* transcript levels during all treatment cycles, leading to a significantly lower relapse rate [104]. An updated analysis of the results observed in the AMLSG 09-09 study confirmed the absence of a significant benefit of GO for EFS and OS [105]. Subgroup analysis showed a benefit of GO in terms of EFS for patients with *FLT3-ITD-WT* and patients with *DNMT3A* mutations [105]. It is important to note that the final results of this trial confirmed that GO administration significantly reduces the cumulative incidence of relapse rate, thus indicating that the addition of GO might reduce the need for salvage therapy [106].

The NCRI AML 29 trial randomized 1475 patients with newly-diagnosed AML or high-risk MDS, with no-adverse cytogenetics, to receive FLG-Ida or DA (Danurubicin plus AraC); 1031 of these patients were also randomized to receive a single or a fractionated dose of GO [107]. Subgroup analysis showed a significant improvement in *NPM1-mut* AML patients treated with FLAG-Ida-GO compared to DA-GO (Os 82% vs. 64% respectively at 3 years); concerning *FLT3-ITD*-mutated patients, a significant OS benefit was observed among *NPM1-mut/FLT3-mut* patients [107].

Several potential targeted therapies against *NPM1-mut* AMLs have been discovered, inhibiting some relevant biochemical properties of the *NPM1* mutant protein, interfering with NPM1 oligomerization or with the abnormal traffic of the NPM1 mutant protein (XPO1 inhibitors), inducing selective NPM1-mutant protein degradation (ATRA/ATO, deguelin, (-)-epigallocatechin-3 gallate), and targeting the integrity of the nucleolar structure (actinomycin D) [107]. The properties of these different drugs and their potential therapeutic implications for *NPM1-mut* AMLs have been recently reviewed [108,109]. Here, the analysis of these drugs was restricted to those in a more advanced stage of clinical evaluation.

Experimental studies have supported a possible efficacy of menin inhibitors in the treatment of *NPM1-mut* AMLs. Histone modifiers MLL1 and DOTL1 control *HOX* gene expression and FLT3 expression in *NPM1-mut* AMLs [110]. Menin-MLL1 targeting inhibited preleukemia cells in a mouse model of *NPM1-mut* AML cells [111] Importantly, menin inhibition synergizes with venetoclax in mediating the inhibition of *NPM1-mut* and *FLT3-mut* AML cells [112]. Similarly, the combination of menin inhibitors with FLT3 inhibitors resulted in an enhanced inhibitory effect on the proliferation and stimulatory induction of apoptosis of primary *FLT3-mut* leukemic blasts [113] and of leukemia cells in a murine model of leukemia promoted by *NPM1-mut* and *FLT3-ITD* [114]. A recent study showed that menin inhibitors synergize with drugs targeting chromatin regulation and DNA damage, as well as with drugs targeting apoptosis and the cell cycle; particularly interesting was the observation of a synergistic interaction between menin inhibitors and ATRA [115].

Fiskus et al. gave an important contribution to the understanding of the biochemical mechanisms through which the menin inhibitor ziftomenib (KO-539) inhibits menin activity: this drug triggers menin protein degradation through the ubiquitin–proteasome system with a consequent marked decline of menin levels and of the expression of menin-dependent genes, such as *BCL1*, *MEIS1*, *FLT3*, *CDK6,* and *MEF2C;* these effects are associated with the induction of leukemic cell differentiation and reduced cell viability [116].

Initial phase I studies have shown a good tolerance and therapeutic efficacy of two menin inhibitors, KO-539 [116] and SNDX-5613 [117]. Recently, the first clinical results of these two menin inhibitors were reported. Issa et al. reported the results of the AUGMENT-101 trial, the first-in-human phase I trial of the menin inhibitor SNDX-5613 (revumenib) in patients with relapsed/refractory AML *KMT2A*-rearranged (46 patients) and *NPM1-mut* (14 patients) patients; these patients were heavily pretreated [118]. In the whole population of 60 enrolled patients, the overall response rate was 53% with a CR rate of 38%, of which 78% were MRD negative; 21% of the *NPM1-mut* AML patients displayed a complete response, with 100% MRD negativity in this responding population [119]. Patients with acquired resistance to menin inhibition displayed somatic mutations in MEN1 at the revumenib–menin interface [120].

The second study, the trial KOMET-001, involved the phase I evaluation of KO-539 (ziftomenib) in adult relapsing/refractory AML patients: in a heavily pre-treated cohort of relapsed/refractory *NPM1-mut* AML patients, an overall response rate of 40% and a complete response rate of 35% were observed [120]. Differentiation syndrome was an adverse event observed in some patients; the occurrence of the differentiation syndrome was associated with an improved response [121]. Future clinical trials will involve the evaluation of the safety and efficacy of ziftomenib in combination with venetoclax plus azacitidine or the 7 + 3 chemotherapy regimen (trial KOMET-007) or the evaluation of revumenib in association with the chemotherapy regimen based on fludarabine and cytarabine (trial AUGMENT-102).

*NPM1-mut* AMLs with *TP53* mutations, as well as other *TP53-mut* myeloid neoplasms, poorly respond to standard treatments. Recent studies are exploring new therapeutic approaches in *TP53-mut* MDS/AML based on immune and nonimmune strategies [122]. Among the nonimmune strategies, some studies are exploring new drugs that restore the activity of the TP53 mutant protein. In fact, some mutations of *TP53*, such as Y220C, determine a destabilization with consequent denaturation and aggregation; the small molecule JC744 interacts with high affinity with the TP53 mutant protein and induces its stabilization [123].

## 15. Conclusions

*NPM1-mut* AMLs represent the largest genetically defined group of AMLs. *NPM1* mutations appear to be secondary events, being virtually absent in CHIP, and occurring after mutations in *DNMT3A*, *IDH1* or *NRAS* during the development of AML. The key role of the *NPM1* gene in these AMLs is clear, but it is equally evident that *NPM1* mutations alone are not able to generate a full leukemic process and must cooperate with other mutant driver genes. The diversity of these oncogenic partners of *NPM1* mutant genes generates a consistent degree of heterogeneity; thus, it was estimated that in these AMLs, there were five oncogenic mutations per patient. This implies that *NPM1-mut* AMLs must be explored by NGS for their mutational profile and consequently stratified in various risk groups. The presence of *TP53* mutations, the triple mutational combination *DNMT3A/NPM1/FLT3-ITD*, and the presence of *MDS*-related co-mutations and of high-risk chromosomic abnormalities are all conditions that shift *NPM1-mut* AML patients into an adverse-risk condition.

## Figures and Tables

**Figure 1 biomedicines-11-01805-f001:**
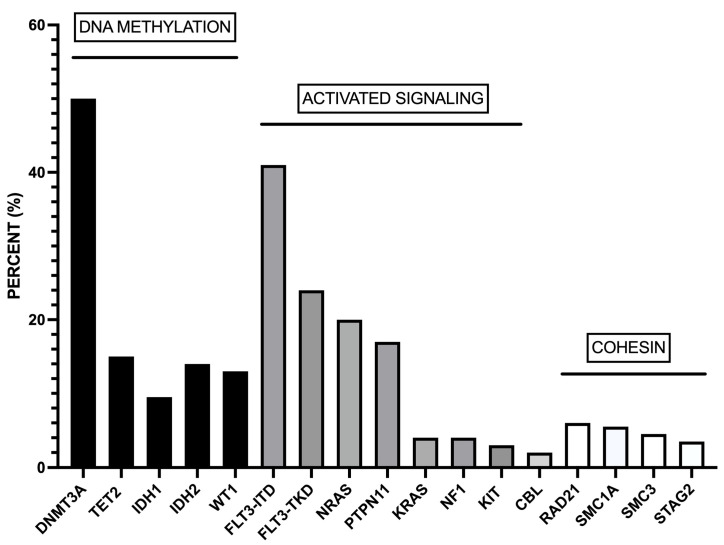
Most recurrent co-mutations observed in adult AML patients. These co-mutations involve genes pertaining to DNA methylation, activated signaling, and the cohesin complex. The data were obtained from Ivey et al. [15].

**Figure 2 biomedicines-11-01805-f002:**
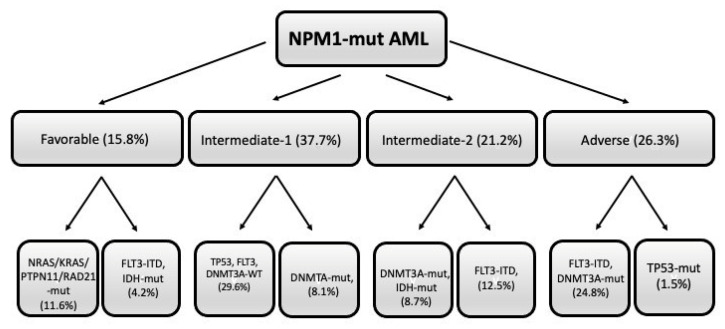
Mutational profile of *NPM-1-mut* AMLs subdivided into four different prognostic groups (favorable, intermediate-1, intermediatye-2, and adverse), each comprising two different subgroups, whose mutational profile is shown. Within parenthesis it is shown the frequency of each subgroup with respect to total *NPM1-mut* AMLs. The figure was freely adapted from data reported by Sanchez et al. [37].

**Figure 3 biomedicines-11-01805-f003:**
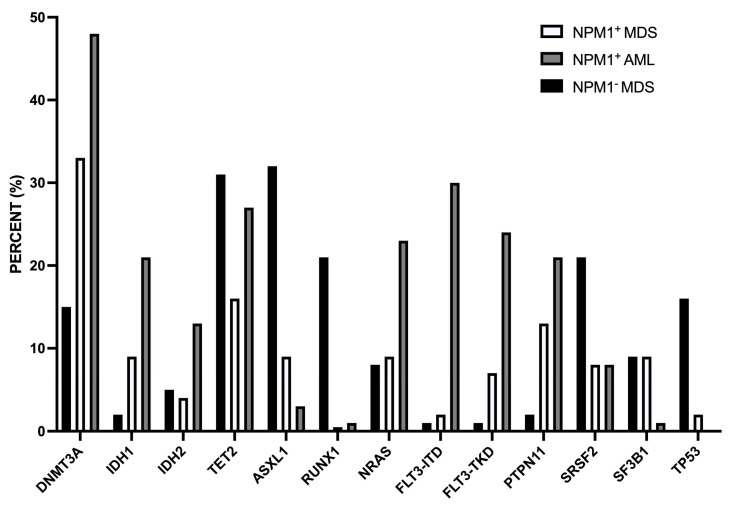
Comparison of the mutational profile observed in MDS *NPM1-mut*, MDS *NPM1-WT,* and AML *NPM1-mut*. The data are reported in Patel et al. 2020 [73].

## Data Availability

No new data were created or analyzed in this study. Data sharing is not applicable to this article.

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
