# Peer review of "Genetic, Phenotypic, and Clinical Heterogeneity of NPM1-Mutant Acute Myeloid Leukemias"

_biomedicines, 2023, doi:10.3390/biomedicines11071805_

Round 1

Reviewer 1 Report

Ugo Testa and colleagues present a quality and well-written review article focused on phenotypic and clinical heterogeneity of NPM1-mutant acute myeloid leukemias.

Authors suggest that recent studies have in part clarified the consistent heterogeneities of these AMLs and have strongly supported the need for an additional stratification aiming to improve the therapeutic response of the different subgroups of NPM1-mut AML patients.

Authors cover such topics as: The mutational landscape of NPM1-mutant AMLs; Cell differentiation heterogeneity of NPM1-mut AMLs; Clonal architecture and clonal evolution of NPM1-mutant AMLs; Prognostic heterogeneity of NPM1-mut AMLs; DNMT3A mutations in NPM1-mut AMLs; FLT3 mutations in NPM1-mut AML patients; IDH1 and IDH2 mutations in NPM1-mut AMLs; Cohesin complex gene mutations in NPM1-mut AMLs; RAS mutations in NPM1-mut AMLs; Myelodysplasia-related alterations in NPM1-mut AMLs; TET2 mutations in IDH1-mut AML; PTPN11 mutations in NPM1-mut AMLs;  Therapy of NPM1-mut AMLs.

Finally, authors conclude that NPM1-mut AMLs represent the largest genetically defined group of AMLs. NPM1 mutations appear to be secondary events, being virtually absent in CHIP, and occurring after mutations in DNMT3A, IDH1 or NRAS during the development of AML. Very importantly they add that the presence of TP53 mutations, the triple mutational combination DNMT3A/NPM1/FLT3-ITD, the presence of MDS-related co-mutations and of high-risk chromosomic abnormalities are all conditions that shift NPM1-mut AML patients into an adverse-risk condition.

Overall, the manuscript is valuable for the scientific community and should be accepted for publication after edits are made.

===========================

Other comments:

1) Please check for typos throughout the manuscript.

2) With regards to mutant TP53 – authors are kindly encouraged to cite the following article that describes the development of novel therapeutic approaches for targeting TP53 mutant cancers.
DOI: 10.1021/acsptsci.2c00164

Author Response

  • Typos have been carefully checked along all the text.
  • Concerning the treatment of TP53-mut NPM1+ AMLs, itw as briefly discussed the papaer suggested by this reviewer (see the end of the section on the therapy of NPM1-mut AMLs).

Reviewer 2 Report

Review of the NPM1mut AML is well-written and comprehensive. It described the presented state of the art of the biology and prognosis of this subgroup of patients. 

Figure 2 which was adapted from Sanchez et al. is not really clear and should be simplified.

Shortening DN AML should be explained in the text (de novo AML?).

Leukemic cellularity (row 321) is not WBC.

row 340 spelled VAT and it is VAF I suppose and so on. 

My suggestion is to shorten the manuscript.

Author Response

  • The Fig.2 was made more clear and itw as simplified, reporting in each box only the most relevant co-mutations to identify NPM1-mut AML subgroups, prognostically relevant.
  • DN AML : itw as now explained the meaning of DN.
  • Row 321, WBC was eliminated.
  • Row 340, VAT was corrected with VAF.
  • We have carefully cheched for typos throughout the manuscript.

Reviewer 3 Report

The review article of Testa et al. is a comprehensive overview of heterogeneity of NPM1c mutated AML with in fact different disease outcomes according to the genetic/phenotypic subgroups.

The review article is very clearly written, complete and pleasant to read even for non-specialists.

I have a minor remark:

It should clearly be stated that NPM1c mutated AML needs no longer 20% of blasts in the bone marrow for diagnosis and the entity of MDS with NPM1c mutation becomes challenging. This might be confusing for readers.

Author Response

The following sentence was added in the section related to NPM1-mut MDS : Since The NPM1-mut AML needs no longer 20% blasts in the bone marrow for diagnosis, the entity of MDS with NPM1-mut becomes challenging.